

# A novel IoT-device management platform for on-the-fly generation of user interface *via* manifest-file addressing heterogeneity

Nayancy Gupta[1], Gourinath Banda[1], Krishna Chaitanya Bommakanti[2] and Venkata Srinivas Kothapalli[3]

[1] Computer Science and Engineering, Indian Institute of Technology Indore, Indore, Madhya Pradesh, India
[2] Research, Adonmo, Hyderabad, Telangana, India
[3] Research, Motorola Mobility, Chicago, IL, United States

## ABSTRACT

The Internet of Things (IoT) is becoming indispensable across various application domains. In the domain of the consumer IoT, many original device manufacturers are coming up with a wide variety of IoT-based products and services catering with a range of applications such as personal-fitness training devices, healthcare devices, to smart-home things, *etc*. There is an accompanying smartphone application, called the IoT control app (ICA) through which such IoT devices are controlled. As of now, a user shall install a separate ICA app for each and every IoT device they own. This is because of the diverse heterogeneity inherent in the IoT domain. The installation of multiple ICAs leads to: memory congestion, steeper battery discharging and increased vulnerability—in smartphones. The diversity in IoT devices can be systematically abstracted away with text written in a manifest file. Based on this manifest file, a user-interface for the IoT-device gets generated on the fly by the ICA. In this article, we propose a manifest-based IoT-device platform including an application-layer protocol, which makes it possible for a single ICA App to control any compliant IoT-device after appropriate authentication. We developed a manifest-grammar for specifying error-free manifest files for different IoT-devices towards a seamless integration between ICA and IoT-devices.

## INTRODUCTION

The Internet of Things (IoT) (*Atzori, Iera & Morabito, 2010*) as a technological paradigm is enabling automation and digital transformation across a wide variety of applications ranging over multiple domains. There are several IoT-application domains (*Pekar et al., 2020; Ibarra-Esquer et al., 2017*), which could be classified under: Consumer, Commercial, Industrial and Infrastructural categories. By 2030, the number of IoT devices worldwide is projected (*Aman et al., 2020*) to cross several hundred billions.

This means most individuals would be accessing several IoT devices. For every IoT-device, their respective device manufacturer provide a smartphone app (application), which is called IoT-device control app (ICA). This ICA is made available on the Google/iOS app-stores, mandating people to install them for each device they own. If one

Corresponding author
Gourinath Banda,
gourinath@iiti.ac.in

owns 50 IoT-devices, then s/he shall install a separate ICAs per device/OEM, thus their phone will have more than 50 ICAs. The installation of multiple ICAs leads to memory congestion and steeper battery discharging in phones. Furthermore, multiple ICA Apps on a device would widen the vulnerability landscape of the phone (*Cui et al., 2020*).

If a single app can handle different IoT devices from diverse vendors then most of the above issues get resolved. With a very generic framework and architecture that abstracts away the diversity through a text-based specification of device capabilities, a single ICA app is sufficient for all IoT-devices. Our solution also includes an ICA and is opensourced.

The contributions of this article is a IoT-device management platform, which includes:

- A manifest-file based approach that homogenises the heterogeneity inherent in the IoT-paradigm as a text file;
- A grammar for the manifest-file that encodes the IoT device capabilities;
- A synergistic application layer protocol and
- One single IoT-device control smartphone app.

The proposed approach empowers an IoT-device to declare its *own identity* and *the functionality* in the form of manifest-file. The basic assumption here is, as the device is carrying its own capability and is able to connect with the user-interface, there is no restriction on the type of the device/domain and the numerous capabilities that a device can offer. This manifest file becomes an integral part of the proposed OneIoT protocol which enables seamless management of IoT-devices by generating on-the-fly graphical user-interface. This article builds on top of our earlier works (*Banda, Chaitanya & Mohan, 2015*; *Banda et al., 2017*), which presented primitive versions of the OneIoT protocol. The most important improvement is the *manifest-grammar* definition. This grammar makes it possible for OEMs and vendors to write error-free manifest-files for their respective IoT-devices. Furthermore, the manifest-file idea is augmented with the concept of *sub-manifest file*. The *main manifest-file* is read-only for the users; while the *sub-manifest file* can be modified to save the user-preferences. These *user-preferences* may include meta information such as: creation of new mode/s, adding the device-location, *etc*. We also illustrate the manifest-grammar's applicability with an example IoT-device.

The rest of the article is organised as follows: related work is summarised in the Related Work section, the Open OneIoT Architecture section details the *Open OneIoT* architecture, the Open OneIoT Implementation section gives the implementation of the protocol including the manifest-file concept and the protocol's architecture, the Experimentation section presents the experimentation, in the Security section we discuss about the security and "Discussion and Conclusion" gives the conclusion & discussion.

## RELATED WORK

Several research efforts, both at individual and consortium levels, have resulted in various *IoT reference architectures* (*Weyrich & Ebert, 2016*; *Shin, 2014*; *Bayer et al., 2004*; *Fernánez, Jaimunk & Thuraisingham, 2023*; *Domínguez-Bolaňo et al., 2022*) across different application domains. Since IoT devices are made up of one or more of: System on Chips

(SoC/s), Microcontroller/s, Field Programmable Gate Arrays (FPGA/s), Sensors, Actuators, Communication controllers and protocols from different vendors, they are inherently heterogeneous. *Atiquzzaman, Noura & Gaedke (2018)*, *Li, Xu & Zhao (2015)*, *Luo et al. (2023)* proposed new frameworks, protocols and standards to achieve seamless interoperability and overcome heterogeneity related restrictions. In the following, we give an overview of various solutions and protocols from different IoT application domains.

Apple provides a platform called HomeKit for connecting the IoT devices. This platform is primarily for devices from Apple. However, it also provides limited support for third-party devices developed using either HomeKit Accessory Protocol (HAP) or Matter protocol (*Apple, 2024*). However, unlike our protocol, there are restrictions (*Sciacco, 2022*) such as: (a) the user-interface can only be on an Apple devices; (b) The cloud entity is specifically iCloud, which is an Apple cloud service and (c) it is only for smart-home domain. The ICA in this approach hardcodes the UI of a specific device associated with that App.

CoAP (*Shelby, Hartke & Bormann, 2014*) is a web transfer protocol suited for communication between IoT-nodes with constrained hardware. It does not propose any ICA application for interaction between the device and their users. Our solution provides a complete IoT device management together with an application layer protocol and an ICA app. On a collaborative front, we can employ CoAP in the place of TCP in our current implementation.

The European Telecommunications Standards Institute (ETSI) has specified a open-standard called oneM2M (*Swetina et al., 2014*), which is applicable for machine-to-machine (M2M) type IoT systems. It is a specification (*ETSI, 2011*); and does not provide a complete open-source software implementation. Furthermore, it does not provide any ICA app as we do in OpenOneIoT; however it defines a possibility for semantic interoperability (*Alaya et al., 2015*).

The "matter" protocol (*Connectivity Standards Alliance, 2022*; *Gennari, 2020*) is yet another project proposed by a consortium that conists of Google, Amazon, Samsung, Silicon labs, *etc*. Unlike our protocol, which is domain independent, the Matter protocol focuses on the smart-home domain. Matter is compatible with Wi-Fi, BLE, Thread (*Unwala, Taqvi & Lu, 2018*), Ethernet, Cellular and 802.15.4-based devices; but does not provide direct compatibility with ZigBee protocol. It is believed to be accessible on a royalty-free basis only by the consortium members. However, our proposed approach works with ZigBee as well but with appropriate brokers in the OS of the IoT-device.

Medical IoT (MIoT) devices (*Alsubaei et al., 2019*; *Dimitrov, 2016*) are at the intersection of Consumer IoT and Industrial IoT, because they include both simple personal devices and sophisticated critical healthcare systems used by medical professionals in hospitals. Again due to the inherent heterogeneity with MIoT devices, they have dedicated ICA apps. However, with our textual manifest file approach, we can have the single ICA to access such MIoT devices as well.

Message Queue Telemetry Transport (MQTT) was released by IBM for lightweight M2M communications. It is an asynchronous publish/subscribe messaging protocol that runs on top of the TCP stack. There is broker containing topics, to which clients can

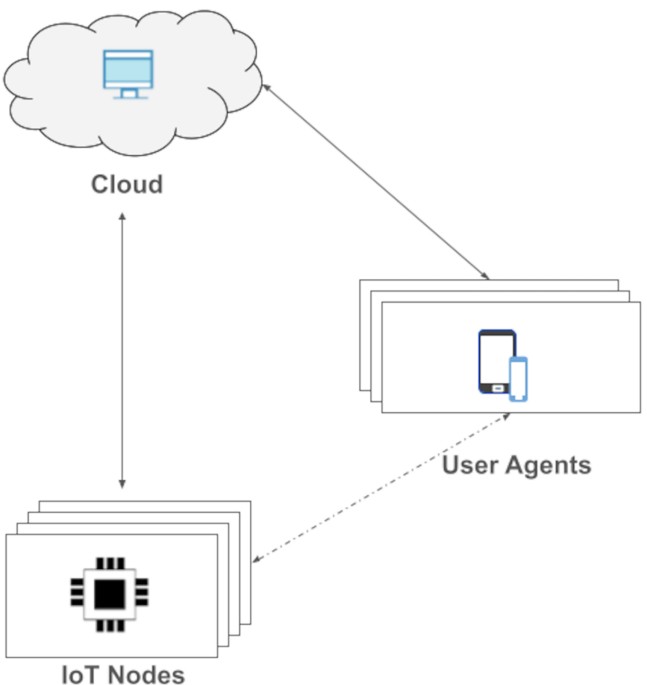

**Figure 1  *Open OneIoT* architecture.**

publish/subscribe (*Karagiannis et al., 2015*; *Yassein, Shatnawi & Al-zoubi, 2016*). There are drawbacks here such as a single point of failure (*Donta et al., 2022*) in broker. While our solution gives a full unified IoT device management; MQTT is just a communication protocol. In MQTT, there is no device management capability. In our protocol, there is a direct communication between client and an IoT device (see Fig. 1). Further, MQTT does not provide any ICA and the new on the fly graphical user generation method.

The Open Connectivity Foundation (OCF) (*Park, 2017*) is a consortium with Samsung, Intel, Cisco, MediaTek, *etc.* Others not in the consortium can access their specification from the GitHub repository (*OCF, 2024*), develop their devices based on it and then getting such devices approved by the consortium. This means a vendor has to present their device specs for approval by consortium, which might not be comfortable due to concerns of loosing competitive advantage. Both the new devices and any upgraded-devices have to get approval. Contrarily, in our solution, any OEM can develop their device specific interface details following the manifest language. The given grammar results in error free manifest files leading to a seamless integration between single ICA and any IoT-device. As the OneIoT ICA app is open-sourced, there is no risk of vendor-lockin as well, because of the standard data exchange format is of JSON-type and the underlying communication is based on the standard TCP protocol.

The Open Mobile Appliance (OMA) (*Brenner & Unmehopa, 2008*) is a consortium including several renowned network companies and mobile communication device manufacturers. It includes names such as: AT&T, Verizon, Samsung, *etc.* OMA's work is targeted for network-operators (*Open Mobile Alliance, 2016*), where it emphasises more on network issues. Our Open OneIoT gives an end-to-end solution through our protocol with

**Table 1 Comparison between different IoT protocols.**

| IoT Platform | Opensource | Compatibility | Connectivity | UI | MobileOS |
|---|---|---|---|---|---|
| Apple Homekit | No | Partial (only Apple complaint devices) | Internet, bluetooth, matter | Hardcoded into ICA | Apple/iOS |
| COAP | Yes | Not applicable (NA) | UDP | NA | Android, iOS |
| Matter | Yes | Partial (only consortium) | Thread, bluetooth | Hardcoded into ICA | Android, iOS |
| MIoT | No | No | No | Hardcoded into ICA | Android, iOS |
| MQTT | Yes | NA | TCP | NA | Android, iOS |
| OCF | Yes | Partial (restricted to the pre populated list) | Internet, zigbee, z-wave | Hardcoded into ICA | Android, iOS |
| Philips Hue | No | Partial (only partners) | Zigbee, bluetooth, Hue bridge | Hardcoded into ICA | Android, iOS |
| Samsung SmartThings | No | Partial (only associated vendors) | Internet, zigbee, bluetooth, matter, z-wave, LAN | Hardcoded into ICA | Android, iOS |
| *Open OneIoT* | Yes | Yes | Independent of the communication media used | Dynamic (generated by the script carried in the IoT device) | Android, iOS |

an included ICA Android app. On the contrary OMA includes a complex architecture, which relies on multiple layers of protocols such as: (a) Generic Bootstrapping Architecture (GBA) for authentication and (b) DM protocols for management.

Philips has developed a connected LED lighting-system called Philips Hue. The lights are controlled with an associated Hue app. Though Philips provides various APIs for developers; there are restrictions on remote API (*Hilbolling et al., 2021*) and reportedly has several security issues (*Morgner, Mattejat & Benenson, 2016*). In Zigbee, as resolution of device IDs to IP addresses is differed, some Zigbee-based bulbs are not compatible. With our OneIoT approach, as we are at the application layer, there remains no restrictions on the type of protocol used for connecting the IoT devices. As our text-based manifest is domain independent, our work is applicable to lighting solutions as well.

SmartThings (*Samsung, 2024*) is a commercial IoT solution from Samsung. Here, there are four types of entities: a hub of IoT-devices, a smartphone app, cloud and the IoT-device/s. The associated SmartThings app is not the app to control the device, rather it lets to install third-party apps, which are the designated ICAs for the respective IoT-devices (*Samsung, 2024*). Not only the users' smartphones get overwhelmed with such additional ICAs, but it also becomes susceptible to cyber-vulnerabilities (*Fernandes et al., 2017*). Samsung has provided multiple ways to integrate third-party devices. For mobile-connected devices, the third-party cloud can only interact with the devices *via* their SmartThings-cloud, which constrains the access to the device.

Table 1 summarises the IoT protocols listed in this section by comparing them along the features of:

- *Opensource*-ness;
- *Compatibility* between different devices with a single ICAs;
- *Connectivity* between the ICA and the device;

- *UI*, whether it is hardcoded into the ICA or it gets generated on the fly by the ICA and
- *mobile/OS* supported by the host.

# OPEN ONEIOT (O2IOT) ARCHITECTURE

A very generic IoT reference architecture could let a single ICA to access, control and monitor any IoT device from any vendor across any application domain. Such an ICA does not need to hard-code the user-interface (UI) for a specific IoT-device; rather can weave the UI on the fly by downloading the manifest-file from the IoT-device. The ICA, once launched for the first time, post device-ownership authentication, downloads a manifest-file from the IoT-device.

## Open OneIoT architecture

We propose a very abstract yet very expressive IoT reference architecture as shown in Fig. 1. Our IoT reference architecture is named Open OneIoT Architecture. The architecture's building blocks, standard protocol and implementation considerations are explained below.

**Definition**: *Open OneIoT Reference Architecture.* The *Open OneIoT* Architecture includes four types of building blocks:

1) IoT-node/s;
2) Remote-host/s (on Cloud) and
3) User Agent/s.

It has an associated application layer protocol (see Fig. 2) which enables user to interact with the IoT device by using a single smartphone App. This protocol also supports dynamic IP address of the device. The remote host always listens on a domain-name (or a static IP-address); IoT device sends a heartbeat packet periodically which helps in getting the IP address of the device.

**Definition**: *IoT node.* IoT node is any IP-enabled embedded device that is uniquely addressable through its IP-address.

This IoT device can be connected and controlled by the smartphone App. The connectivity medium could be internet or other means. The dotted line depicts non-internet based connectivity. An IoT-node shares its IP-address with the Cloud node, which forms the basis for the app to connect with it. Each IoT-device is preloaded by its manufacturer with a small manifest-file in its flash memory. This file is utilised for advertising the device's functionalities.

As the proposed solution is in the application layer, it is independent from the network stack, which is implemented in the operating system on the device. So it natively works on HTTP; if there are brokers installed on the OS then it would work for CoAP, MQTT, *etc.* This confirms to the OS implemented TCP/IP stack. Hence our protocol has conforms to the stack as shown in Fig. 2.

**Definition**: *Remote-host.* Remote-host is the server belonging to the OEM that provides the service of user-device ownership information. This runs in the cloud. Cloud always

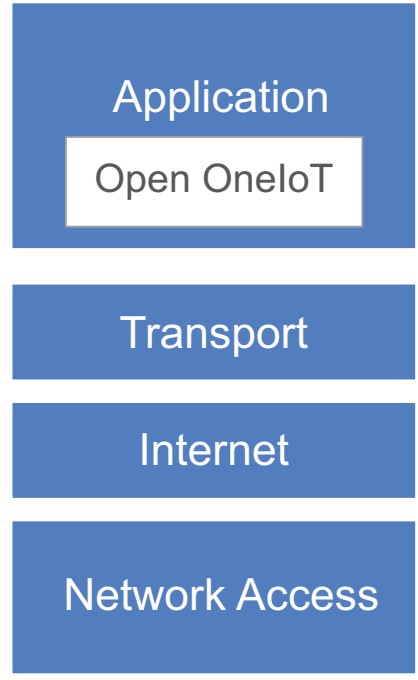

**Figure 2 Protocol stack.**

listens at a hostname (or a fixed IP address). This server is the point of communication to get the the IP address of the device in case of IP changes. It also manages the user-device mapping which holds the information about IoT devices affiliated to the user.

**Definition**: *User agents*. User agent is simply a client interface through which an IoT-device is accessed by user. Basically it is a device on which ICA runs. For example, a smartphone, tablet, laptop, *etc.* With the help of user-agent, one can control the IoT devices.

The interaction between these entities in the architecture are defined in the protocol's phases as explained later under "Open OneIoT Protocol".

## Open OneIoT protocol

In the IoT paradigm, the key feature is that an IoT-device can be uniquely accessed by its IP address. Once an IoT-device gets an IP-address, an IoT protocol defines a standard sequence of steps or phases that brings the device's capabilities delivered to its user. There are several IoT protocols applicable at different layers. Our Open OneIoT protocol is at the application layer. There are several steps in the protocol which are systematically anchored through a small *manifest* file. The Open OneIoT protocol includes the following phases in that sequence:

1) Discovery;
2) Registration;
3) Probe/Advertisement;

| Protocol Name &Ver | Encryption Type |
|---|---|
| Payload | |

**Figure 3  Request/response message structure.**

| Protocol Name | Operation name | Arguments |
|---|---|---|

**Figure 4  Protocol request payload.**

Outputs

**Figure 5  Protocol response payload.**

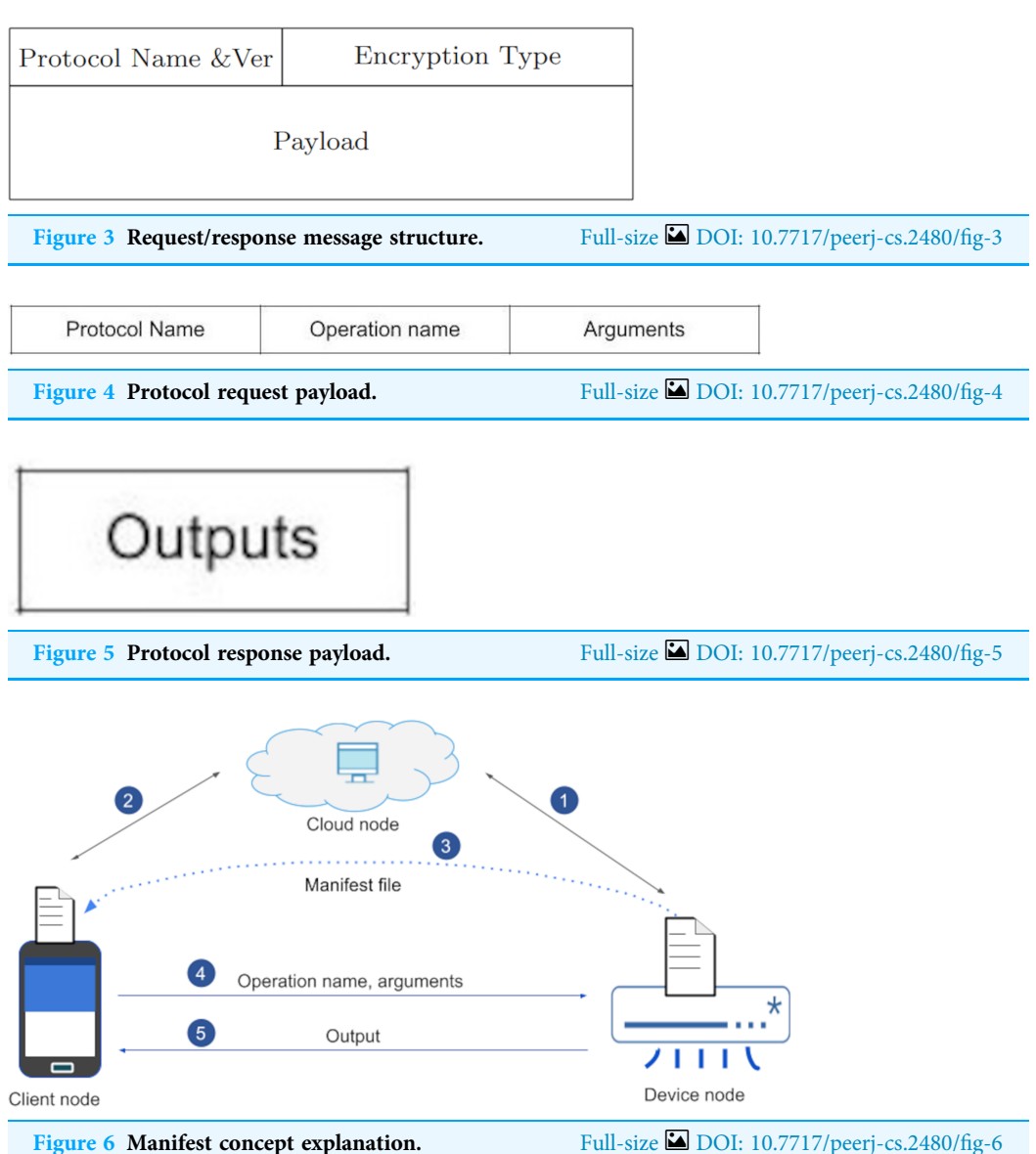

**Figure 6  Manifest concept explanation.**

4) Invocation;

5) Configuration.

The protocol message format includes header and payload. Message structure is described in Fig. 3. Every phase in the protocol includes a well defined payload structure. The payload structures transmitted during the *Request* and *Response*-phases is given in Figs. 4 and 5, respectively. For example, in the case of a PROBE-request, the *protocol name* entry will be *PROBE*; while the *operation name* and *arguments* entries will be empty. In Fig. 6, Point 4 corresponds to the protocol's request payload; while Point 5 in corresponds to the protocol *Response*-payload.

The security concerns can be addressed *via* techniques such as: encryption, authentication, firewalls on the terminal points, rate limiting, and other techniques as

surveyed in *Lee (2020)*. We employ the encryption technique in our protocol (*Banda, Chaitanya & Mohan, 2015*). In all phases, the messages are encrypted. There are several advanced techniques for mitigating different kinds of attacks as reported in *Tripathi & Hubballi (2022)*. Since IoT-device hardware is resource constrained, by using state-of-the-art D/DoS filter services available on cloud these attacks can be averted. The idea is that all the inbound traffic to the IoT-device would always be routed *via* this cloud-filter.

The Discovery-phase makes it possible for discovering the IoT-device by performing a local scan *via* WiFi, Ethernet over LAN, Bluetooth and other short ranged protocols.

The Registration-phase involves the discovery of the IoT device *via* IP-based communication. User-agent shares the IoT device credentials with the remote host on the cloud. After verification, the remote host shares the IP address of the IoT device with the ICA hence IoT device gets connected with ICA.

In the Probe/Advertisement-phase, the IoT device sends the manifest file to the ICA, which by interpreting the manifest file generates the graphical user interface on the fly. The *manifest-file* pronounces the functionalities for the associate IoT-device in a standard manner compliant with the manifest grammar that is presented later.

There are two scenarios for the joining of new nodes, it could be that the new node is: (i) an IoT-device node and (ii) a new user-agent node. The OneIoT protocol includes solutions for both these scenarios through its: (a) Discovery and Advertisement phase, where new IoT-device nodes gets onboarded and (b) Registration phase, which lets a new user-agent paired with the IoT-device.

Now, in the Invocation phase, the user invokes/requests any operation on the IoT device by using the GUI generated in the probe phase.

The generic configurations of a device is addressed in the Configuration-phase of the protocol. For example *tick rates, security parameters* and *other protocol variables*. These changes can be helpful in improving the performance of the system.

The Sequence diagram corresponding to protocol's Registration, Probe and Invoke phases are shown in an integrated fashion in Fig. 7. An elaborate explanation on these phases can be found in *Banda, Chaitanya & Mohan (2015)* and *Banda et al. (2017)*, where OneIoT protocol was reported. *Open OneIoT* suite retains the OneIoT protocol's phases.

## Steps and illustration

The IoT-device manufacturer will generate the manifest-file (steps for generation is defined in "Illustration of MMF Generation") and place it in the IoT-device. It is in the non-volatile flash memory of the microcontroller. When a user purchases the IoT device, it is already equipped with the manifest file provided by its OEM. Additionally, manufacturer also provides the first time device registration credentials.

Once the device starts up, it will automatically connect to the cloud (see point 1 in Fig. 6). The user will download and install the IoT-device Control App (ICA) on their smartphone (client node). This ICA when launched for the first time lets choose appropriate options and pair an IoT-device. After pairing has been done, next step is registration.

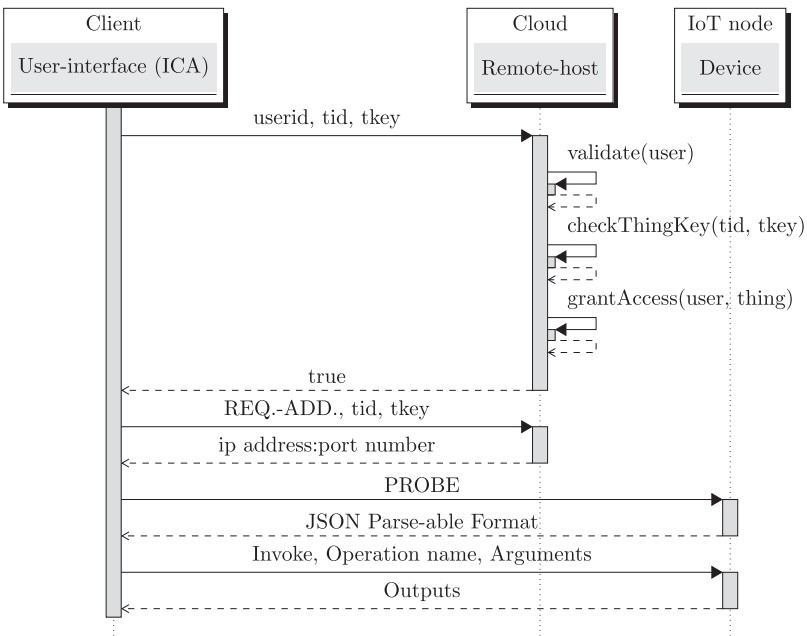

**Figure 7** Protocol sequence diagram: registration, probe and invoke phases.

For registration purpose, a user will fetch the device's and OEM's details (see "Experimentation" in Screenshot 3 & 4) which the device's OEM has already shipped to the user along with the IoT-device. These credentials after keyed-in on the client-node and will get authenticated by the cloud-node. Once authenticated, the cloud-node will share the IP-address of the device (see point 2 in Fig. 6). Once Registration succeeds, manifest-file gets transferred (see point 3 in Fig. 6) to the ICA and the ICA generates the UI (see "Experimentation" in Screenshot 5). Now, user can invoke any operation on the device by selecting the associated options in the GUI (see point 4 in Fig. 6). Once the operations are invoked on the ICA App, the corresponding IoT-device will receive a message, which is checked for its validity and the requested-operation is carried out and accordingly the status is updated to the ICA (see point 5 in Fig. 6). The "Experimentation" section details more on these steps.

## OPEN ONEIOT IMPLEMENTATION

In this section, we elaborate on the realisation of Open OneIoT protocol with a focus on the implementation of the manifest file concept. Since this manifest-file is transferred between devices post appropriate authorisation, we chose to specify it in a JSON file. JSON (*Zhang et al., 2017*) is a popular Data-interchange-format that has gain popularity in IoT paradigm (*Hou et al., 2017*). We defined a grammar for specifying the manifest in this format. Though JSON and XML are popular format, we chose JSON over XML, for three reasons of (*Wehner, Piberger & Göhringer, 2014*; *Rasool et al., 2019*): (i) JSON is lightweight as it has no markup overhead as that of XML; (ii) for the same data JSON-encoding is much more compact in size as compared to that of XML-encoding and (iii) that XML-file need more processing power than that with JSON-file.

## IoT-device manifest file

In our Open OneIoT architecture, the most important concept is the ability to declare the capabilities of the IoT-device in the Main Manifest File (MMF). It includes details about the device's functionality, particularly, the controllability related details. There are two kinds of manifest files: *Main-manifest-file* (MMF) and *Sub-manifest-file* (SMF). The MMF is mandatory; while the SMF is optional.

A manifest-file consists of two types of sections: (i) **Required section/s**: These sections are essential and are interpreted by the associated ICA to reveal the minimum required/basic behaviour of that IoT-device. Of these sections, it is mandatory that the main manifest file shall include either a *sensor* (see "Main Manifest File") or *actuator* (see "Main Manifest File") and (ii) **Optional section/s**: These are sections such as *device, oem, location* and *mode*, which are optional sections, for these might not be the essential functionalities offered by a device.

### Main manifest file

The *main manifest-file* (MMF) captures the list of all predefined-set of functionalities of an IoT-device. This file is must for a device and it can not be modified by a user. It consists of sections, such as:

– Device;
– OEM;
– Sensor;
– Actuator and
– Mode.

**Device** This section provides information about the device such as: device-name, domain of the device (smart-home, medical, *etc.*), installation-type, *etc.*

**OEM** The OEM-section provides information about the manufacturer of the device. This will be helpful in identifying the OEM-server. It includes the URL of the server, with a Hostname/IP-address along with a port-number.

**Sensor** A device can have various sensors attached to it. All the sensors of the device will be listed under this section. The Sensor's value is read-only value and a client can not make any changes to the value. For example, smart-wearables might have sensors (for example: Accelerometer, Pulse-oximeter, Temperature-sensor, Proximity-sensor, Pulse-sensor, *etc.*), which collect the data about a person's vitals and send this data to the ICA running on the smartphone (client node). However, if it is on the actuator version, then the SET operation with actuator name is invoked. This is explained under the Actuator section next.

To get a sensor's current reading value, the 'invoke' operation can be utilised (see Fig. 8). Here, the *get method* of *Invoke type* on the wire will instruct the device to send the sensor's value as an *Invocation response*.

**Actuator:** Actuator is a component in the IoT-device that can be controlled by a user. All such actuators are defined under this section of main manifest file. Here, we have the

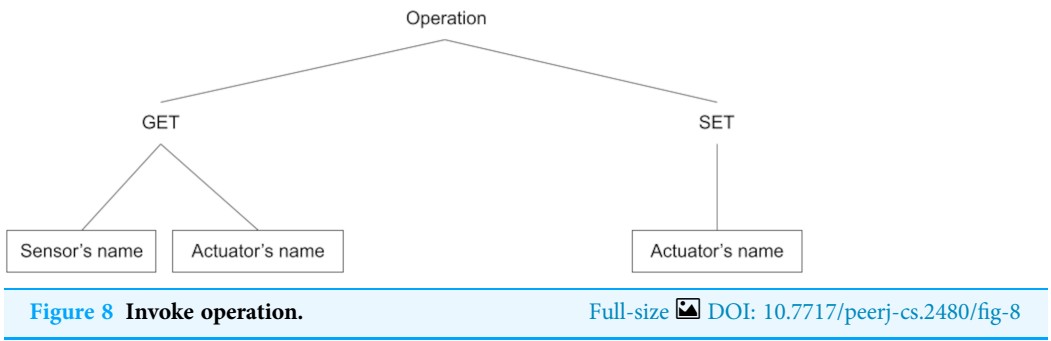

**Figure 8 Invoke operation.**

definition of the actuators like its description and data-type. This information is helpful in generating the GUI for the actuators available in the IoT-device. Later, this GUI is accessed by user for invoking any operation on the IoT device.

For example, in an *RGB LED* we have colour and brightness as actuators. For controlling, we can have *set* and *get* both types of *invoke methods* as illustrated in Fig. 8. To get the current value of the actuator, GET method will be utilised and for changing the value, *Invoke type* will be SET and device will set the actuator's value given in the payload and send an *Invocation response* to the user-interface.

**Mode:** Mode is the group of actuators having some fixed predefined or user-defined values. *Predefined-modes* are saved in main manifest file and *user-defined modes* are saved in sub manifest file. OEM provides the predefined-modes of a device. A user can invoke the mode by selecting it. Sometimes, having pre-defined values for a group of actuator is beneficial as this will save time for the user and any chance of assigning wrong value to actuator can be avoided completely. For example, in LED bulb, the "night" mode can be helpful in dimming the light, this will simultaneously set the value of *brightness actuator* to *low* and *colour actuator* to *yellow*. This "Mode" is an optional section.

### Sub manifest file

Sub manifest file (SMF) will be created by user when s/he adds any new functionality or information of the device and assign a chosen value. When user *resets* the device (either *via* app or hardware button), this SMF gets deleted. Even when reset, the MMF will remain in the device. There can only be one SMF on a device. The contents of the sub manifest file are as follows:

– Mode and
– Location.

**Mode:** Mode is the group of actuators. It becomes easier when the device has multiple actuators. These modes are saved in SMF. A user can create the mode by selecting the desired actuators and choosing their setting-values.

**Location:** A device can be mounted in a fixed location (for example: AC, fan, lightbulb) or can be carried anywhere (for example: speaker, wearables, blood glucose meter *etc.*). The location information is fetched by the user. In the ICA, devices can be grouped on the basis of their location and this information helps in taking the decision based on the location of

the device. This can be helpful in managing the device smartly. For example, in a kitchen, there can be one fan, one water heater and two LED bulbs installed; one can simply switch off all the devices by selecting *switch off* all the appliances in the kitchen location.

## Manifest grammar

The manifest-file shall be error-free, because it is shipped with the IoT-device by the manufacturer and gets processed by the ICA on the user's smartphone. The ICA generates the UI on the fly by processing this file. We have defined this Manifest grammar that makes possible for error free manifest file generation by the device manufacturers. The grammar explains each section of the given manifest file. We can construct the manifest of the device with this grammar.

We define the manifest grammar as a 4-tuple G = (V,T,P,S), where:

```
V ∩ T = {Φ};
V: Finite non-empty set of non-terminal symbols;
T: Finite set of terminal symbols;
P: Finite non-empty set of production rules and
S: Start symbol set.

V->{S, START, OPTIONAL, SENSOR, ACTUATOR, DEVICE, OEM, MODE, LOCATION,
D, NAME, DOMAIN, WARRANTY, INSTALLATION, SW_VERSION, HW_VERSION, O,
SOCKET, M, NAME_VAL, Z, DATA, ACTUATOR_INFO, VALUE, UNIT_VAL, AREA, X,
DESC, SENSOR_TYPE, STRUCT_X, BATTERY, GRAPH, DATA_TYPE, PIE, BAR,
GRAPH_TYPE, CORD1, CORD2, BOOLEAN, NUMERIC, STRING, IMAGE, AUDIO, VIDEO,
MAP, COLOR, DATE, TIME, RANGE, TUPLE, UNIT, R, RVAL, MIN, MAX, STEP,
UNIT_VAL, OPERATION, EXCEPT, TVAL, OPTION, OVAL, SIZE, IMAGE_ATTRIB,
LENGTH, WIDTH, SRC, IMAGE_TYPE, PLAYER_TYPE, PLAYER_ATTRIB, LOOP, MUTE,
ACTUATOR, Y, ACTUATOR_TYPE, STRUCT_Y }

T->{num, pos_num, str, bool, array, jpeg, png, gif, raw, time, [, ],
{, },:, ','}

S->{S}

P is defined as below:
S ->{START}
START -> OPTIONAL,START |SENSOR | ACTUATOR | SENSOR,ACTUATOR
OPTIONAL -> DEVICE | OEM | MODE | LOCATION

DEVICE ->:{D}
D -> NAME | DOMAIN | INSTALLATION | WARRANTY | SW_VERSION | HW_VERSION | |
D,D NAME, DOMAIN, INSTALLATION, WARRANTY, SW_VERSION, HW_VERSION ->:str

OEM ->:{O}
O -> NAME | SOCKET | NAME,SOCKET
SOCKET ->:str:num
```

```
MODE ->:{M}
M -> M,M | NAME_VAL:{Z}
NAME_VAL -> str
Z -> ACTUATOR_INFO | ACTUATOR_INFO,Z
ACTUATOR_INFO -> NAME_VAL: DATA
DATA -> [VALUE] | [VALUE,UNIT_VAL]
VALUE -> str | num | array
UNIT_VAL -> str

LOCATION ->:[AREA]
AREA -> str | AREA,AREA

SENSOR ->:{X}
X -> X,X | NAME_VAL:{DESC, SENSOR_TYPE} | NAME_VAL:{SENSOR_TYPE}
DESC ->:str
SENSOR_TYPE -> STRUCT_X | BATTERY | GRAPH | DATA_TYPE | {}
STRUCT_X ->:{X, X}
BATTERY ->:{}
GRAPH ->:{GRAPH_TYPE}
GRAPH_TYPE -> PIE | BAR
PIE, BAR ->:[CORD1,CORD2]
CORD1, CORD2 -> num | str | bool | time

DATA_TYPE -> BOOLEAN | NUMERIC | STRING | IMAGE | AUDIO | VIDEO | MAP |
COLOR | DATE | TIME

BOOLEAN ->:[str, str]

NUMERIC ->:{RANGE} |:{TUPLE, UNIT}
RANGE ->:[R,R] |:RVAL
R -> RVAL | R,R
RVAL -> [MIN,MAX,STEP,OPERATION] |[MIN,MAX,STEP,OPERATION,UNIT_VAL]
|[MIN,MAX,STEP,OPERATION,UNIT_VAL,EXCEPT]
MIN,MAX -> num
STEP -> pos_num
UNIT_VAL -> str
OPERATION -> + | * | -1
EXCEPT -> [TUPLE] | [MIN,MAX,STEP,OPERATION]
TUPLE ->:[TVAL]
TVAL -> TVAL,TVAL | num
UNIT ->:str

STRING ->:{OPTION}
OPTION ->:[OVAL]
OVAL -> str | OVAL,OVAL
```

| Algorithm 1 | Algorithm for traversing the tree and generating the JSON code |
|---|---|

1: Start

2: Go to the start symbol and start traversing the tree from left

3: For each terminal node repeat steps from 4 to 7

4: **if** (terminal == ":" and isleftmostchild)

5:      print("parent node")

6: **end if**

7: print(terminal)

8: Stop

```
IMAGE ->:{SIZE, IMAGE_ATTRIB} |:{SIZE} |:{IMAGE_ATTRIB} |:{}
SIZE ->:[LENGTH, WIDTH, UNIT_VAL]
LENGTH, WIDTH -> pos_num
IMAGE_ATTRIB ->:[SRC, IMAGE_TYPE]
SRC -> str
IMAGE_TYPE -> jpeg | png | gif | raw

AUDIO ->:{SIZE, PLAYER_ATTRIB} |:{SIZE} |:{PLAYER_ATTRIB} |:{}
PLAYER_ATTRIB ->:[SRC, PLAYER_TYPE, LOOP, MUTE, ACTUATOR, AUTOPLAY]
PLAYER_TYPE -> mp4, mp3, ogg, mov, wmv, avi, mpeg, webm
LOOP, MUTE, ACTUATOR, AUTOPLAY -> bool

VIDEO ->:{SIZE, PLAYER_ATTRIB} |:{SIZE} |:{PLAYER_ATTRIB} |:{}

MAP ->:{SIZE} |:{}

COLOR ->:{}

DATE ->:{}

TIME ->:{}

ACTUATOR ->:{Y}
Y -> Y,Y | NAME_VAL:{DESC, ACTUATOR_TYPE} | NAME_VAL:{ACTUATOR_TYPE}
ACTUATOR_TYPE -> DATA_TYPE | STRUCT_Y
STRUCT_Y ->:{Y, Y}
```

## Algorithm

The grammar defined earlier in "Manifest Grammar" forms the basis for the MMF-generation-tool. The algorithm for traversing the parse tree generated by applying this grammar is given in Algorithm 1. Here, the 'isleftmostchild' is the outer left child of a node. We have explained the generation of the JSON-code by applying this algorithm in "Illustration of MMF Generation".

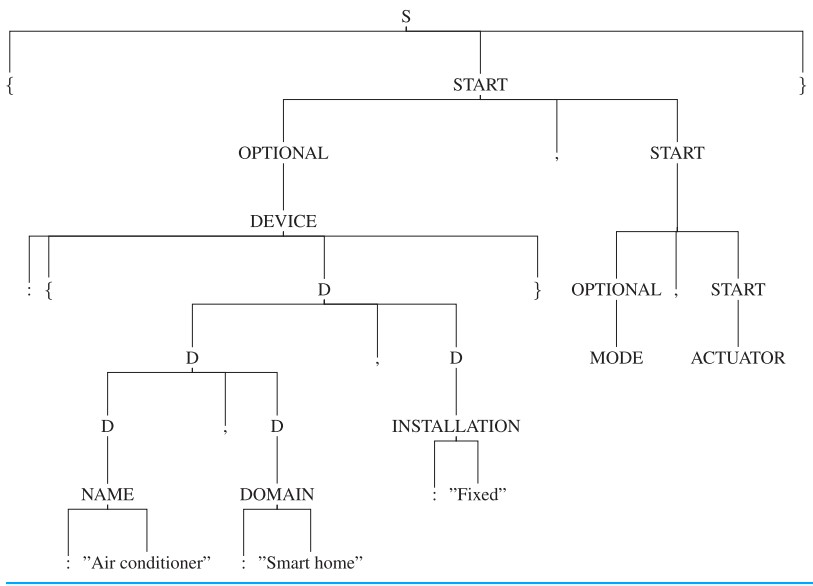

**Figure 9 Tree.**

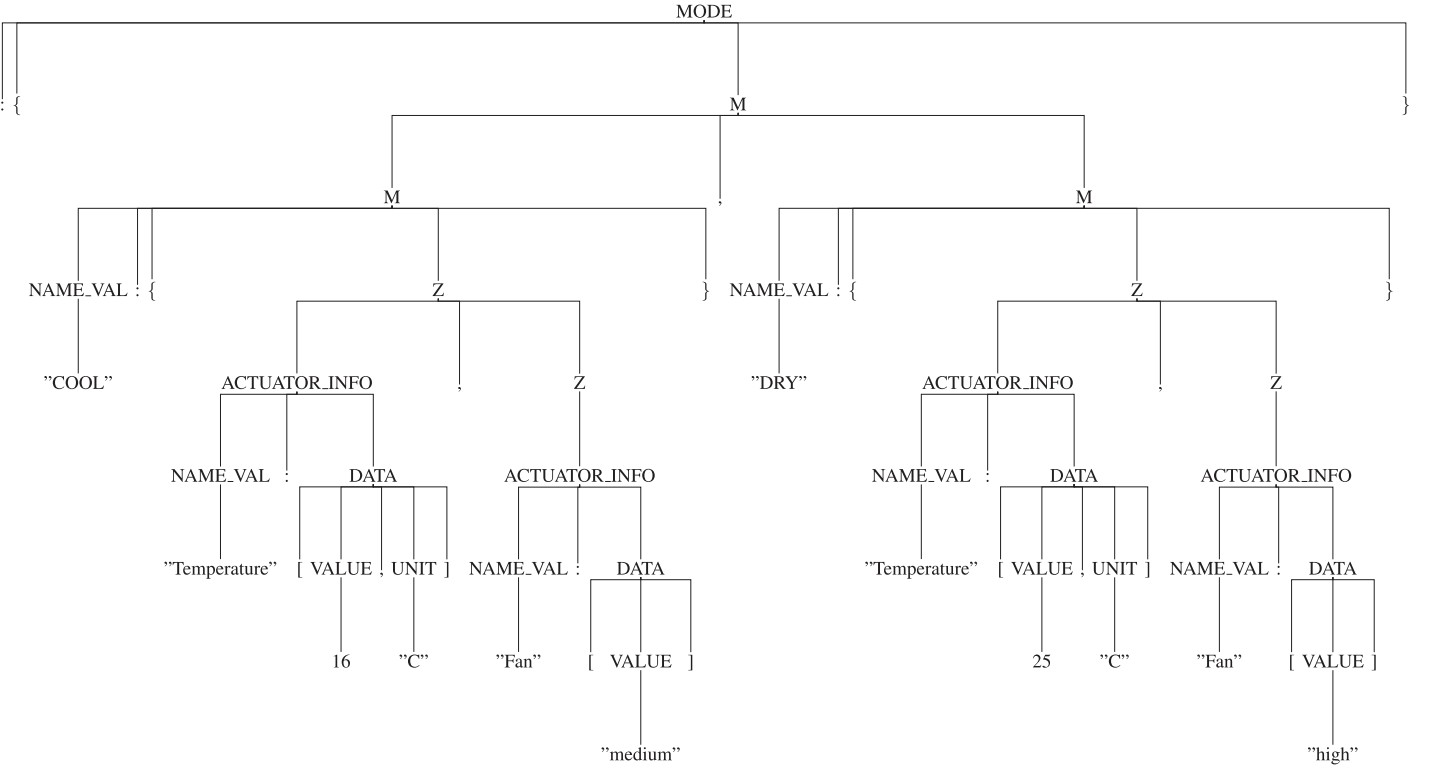

**Figure 10 Mode subtree.**

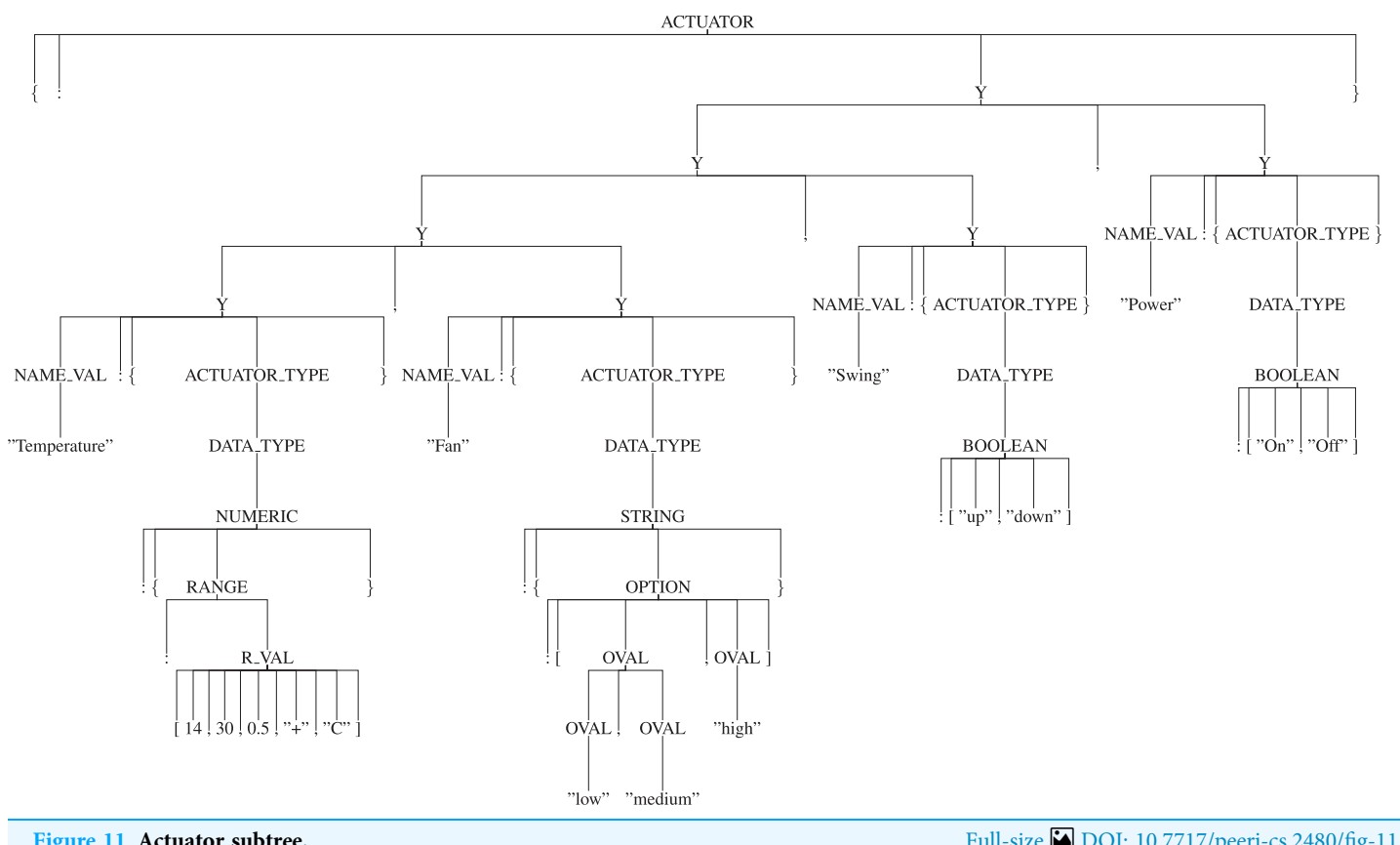

**Figure 11 Actuator subtree.**               

## Illustration of MMF generation

This section explains the steps involved in generating the manifest file for a device. This is illustrated by taking an air conditioner (AC) as the running example. The considered AC unit has features such as: temperature, fan-speed, swing, *etc*. We have taken the reference of Havells, LG, Samsung and Whirlpool AC models and their associated mobile-applications.[1] A manifest file for this air conditioner has been written as per the grammar defined earlier. Below are the steps given for generating the MMF.

    **Choice of sections and attributes:** Firstly, the OEM needs to choose what sections to be included in the main manifest file Section. The sections were detailed under "IoT-Device Manifest File". Next, chosen are the applicable attributes for each section. These attributes are defined in the "Manifest Grammar". For example, say required sections are, namely: *device*, *actuator* and *mode*. Furthermore, under the device-section, the attributes are: *name*, *domain* and *installation*. Under the *actuator section*, we can append the hardware (actuator) which can be controlled by a user. For the AC, actuators: *temperature*, *fan*, *swing*, *power-on/off*, *etc*. can be controlled by the user. We have defined predefined modes such as: *Cool* and *Dry*. As stated earlier in "Main Manifest File (MMF)", modes are the combination of actuators. The *Predefined modes* are already populated with appropriate fixed-values. For example, here cool-mode is pre-populated with *temperature* as 16 °C and

[1] These apps are downloadable *via* app-stores corresponding to Android and iOS.

```
1   {
2           "DEVICE" : {
3                   "NAME":"Air conditioner",
4                   "DOMAIN":"Smart home",
5                   "INSTALLATION":"Fixed"
6               },
7           "MODE" : {
8                   "COOL" : {"Temperature" : [16,"C"],
9                              "Fan" : ["medium"]
10                          },
11                  "DRY" : {"Temperature" : [25,"C"],
12                             "Fan" : ["high"]
13                          }
14              },
15          "ACTUATOR" : {
16                  "Temperature" : {
17                          "NUMERIC":{
18                                  "RANGE":[14,30,0.5,"+","C"]
19                          }
20                      },
21                  "Fan" : {
22                          "STRING":{"OPTION":["low","medium","high"]}
23                      },
24                  "Swing" : {
25                          "BOOLEAN":["up","down"]
26                      },
27                  "Power" : {
28                          "BOOLEAN":["on","off"]
29                      }
30              }
31      }
```

**Figure 12 Main manifest file (MMF) example.**       

*fan-speed* as *medium*. These attributes are essential details that will be shown on the user-interface *i.e.*, in the ICA.

**Usage of the appropriate production rules for generating the parse tree:** By using the appropriate production rules defined in the "Manifest Grammar", parse-tree gets created. In this AC example, as per the defined production rules, 'S' is the start symbol *i.e.*, the root of the tree, it will be reduced to '{START}' by using the production rule ʹS -> {START}ʹ . Here, '{' and '}' are the terminals and 'START' is the non-terminal. So, further 'START' will be reduced by using the appropriate production rule and it will continue until all the leaf nodes reduce to terminals. Thus, it will create a parse tree as shown in Fig. 9. Figures 10 and 11 are the sub-trees of the tree generated in Fig. 9.

**Applying Algorithm 1 to traverse the parse tree:** After the parse tree has been created, the tree can be traversed as per the steps given in the Algorithm 1. First, start from the root-node (*i.e.*, start symbol) and then consider traversing from left, a for-loop will start in Step

3. As per the Step 4, the leftmost node is '{' (*i.e.*, terminal) and not equals to ':', hence Step 7 will be executed and it will print the '{'. Now, again go to Step 4 and check for the next terminal, the next terminal node is ':' and 'isleftmostchild' is also true, hence Step 5 will be executed and 'DEVICE' (*i.e.*, parent node of ':') will be printed and then as per Step 7 ':' will be printed. Again the Step 4 will be executed and this will continue until all the terminals have been evaluated. This will output the JSON-code as shown below:

```
{"DEVICE":{"NAME":"Air conditioner","DOMAIN":"Smart home",
"INSTALLATION":"Fixed"},"MODE"
 :{"COOL":{"Temperature":[16,"C"],"Fan":["medium"]},"DRY":
{"Temperature":[25,"C"],"Fan":["high"]}},"ACTUATOR"
 :{"Temperature":{"NUMERIC":{"RANGE":[14,30,0.5,"+","C"]}},"Fan":
{"STRING":{"OPTION":["low","medium","high"]
 }},"Swing":{"BOOLEAN":["up","down"]},"Power":{"BOOLEAN":
["on","off"]}}}.
```

**Saving the generated code as a JSON-file:** After parsing the tree through the algorithm we get the output produced, which is a JSON-code. Figure 12 shows it in readable format. Typically, for every controllable feature of an IoT-device, there would be an entry in the JSON-file. Each of such entries would be of length ranging from 30 to 50 characters (*i.e.*, max 50 bytes). For the AC-example, the manifest-file is of 375 bytes size. This can be easily stored in the flash-memory of microcontroller corresponding to the IoT device. The transfer of this small manifest file would be almost instantaneous on a high bandwidth connection. On a 1 Mbps line, this file would take less than 0.4 ms for its transmission. Once a manifest-file (in JSON-format) is received, the ICA parses to generate the user-interface on the fly. On user's smartphones, this parsing and UI-generation would complete in a fraction of a second. Other competing formats for encoding manifest could be XML. In the case of manifest being in XML-format, then a server is hosted on the IoT-device and browser is used to access the device, then it would have resulted in a very heavy memory-footprint scenario. Our Open OneIoT's IoT-device side Python script is less than 2 KB; while a smallest server would be in the order of several MBs. On the client-side, our Open OneIoT ICA is of 34.5 MB size; while most browsers are larger than 100 MBs at least. However, reading (resp. writing) the sensors (resp. actuators) on the IoT-device through server would be difficult and highly resource intensive, as server runs as a separate process on the IoT-device. To overcome all these drawbacks associated with XML-formal, in our implementation we chose JSON-format for our manifest file.

## EXPERIMENTATION

In the experimentation, we first did simulation and then we implemented them on corresponding hardware.

We have simulated the behaviour of these three nodes by writing three different processes on the same machine in Python language. We have implemented a client-node, device-node and cloud-node. Here client node (see Fig. 13) is a user-interface for the end-

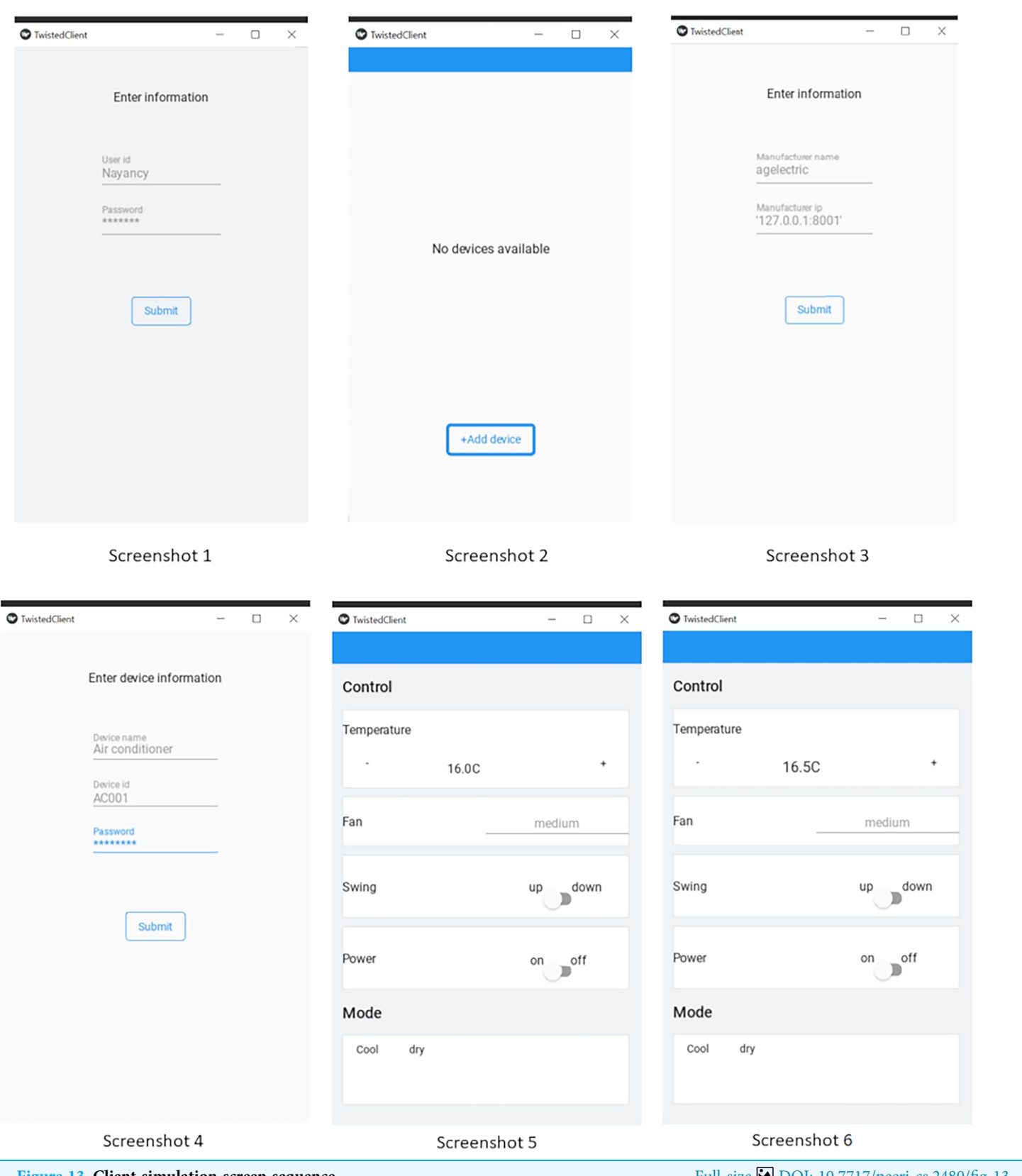

**Figure 13 Client simulation screen sequence.**

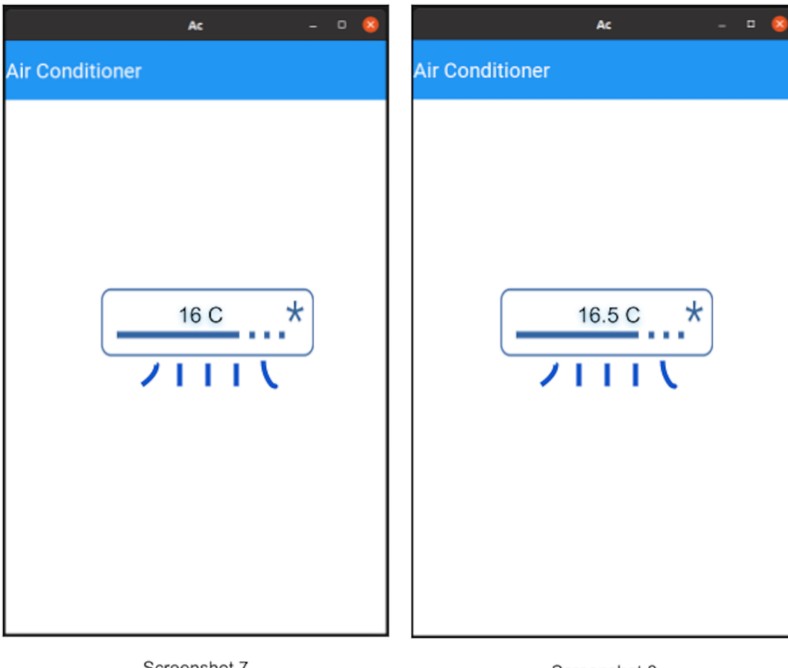

**Figure 14 Air conditioner simulation.**

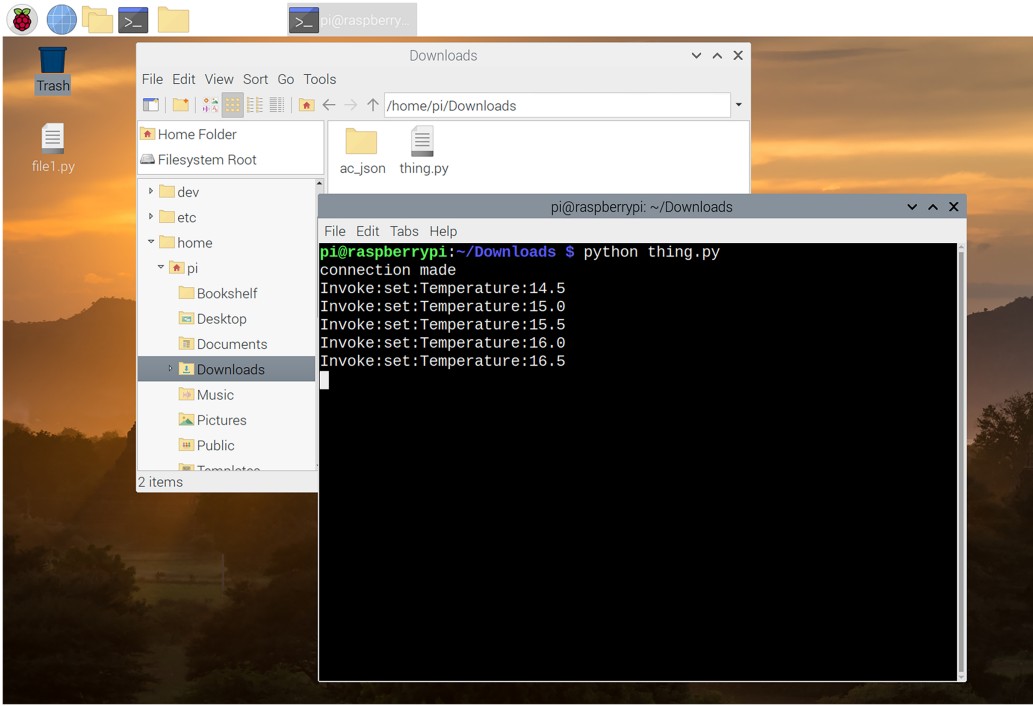

**Figure 15 Raspberry IoT node.**

consumer; whereas IoT-node (see Fig. 14) is the IoT-device and the Cloud-node is for the OEM's server, where remote-host is running. We used *Twisted framework* (*Kinder, 2005*) for network programming and by using *Kivy framework* (*Phillips, 2014*) for developing

graphical user interface (GUI). The Cloud-node is also written in Python using Twisted-framework. All the three processes were run on a single computer having an i7 Processor with 16 GB RAM running an Ubuntu Linux Operating System.

After a successful simulation setup, we proceeded for a real setup. Towards this, we have developed an Android-application, called *OOneIoT-App*, which is the ICA. This App (apk) was installed on a Samsung Galaxy Android smartphone. The IoT-device node was implemented with a Raspberry Pi 400 board (*McManus & Cook, 2021*) (see Fig. 15). For the Cloud-node (*i.e.*, OEM-server), we installed a server (a process in Python) on an Intel i7 processor based Ubuntu system. Thus, the three nodes, *i.e.*, thing-node, user-agent-node and the Remote-host were made to run on three different-machines, which were connected *via* internet. The different phases of the protocol in action are explained in the following together with their corresponding ICA-screenshots.

**User login:** Firstly, Screenshot 1 shown at the leftmost corner in Fig. 13 corresponds to the login step. Here you are seeing the landing screen after the ICA App has been installed on the smartphone. The user has to *login* into the ICA by providing a *Unique user-id*. The user-credentials can be the Email-ID and/or mobile-number with an associated password. Once these credentials are keyed in on the ICA, the Remote-host on cloud will uniquely identify and authenticate the client. After a successful *Log-in*, a list of associated devices will be displayed (which is saved in user-interface). For the *first-time* login, this list will be empty, for no devices have been added by the client. This can be seen in Screenshot 2 of Fig. 13.

**Registration-phase:** Here, we illustrate the Registration-phase in the protocol. The Client *via* ICA can add any device using *Discovery* or *Registration phase* of the protocol. The Client provides the IoT-node information (such as device-id and password), which are authenticated by the remote-host. This can be seen in Screenshot 3 & 4 of Fig. 13. In Screenshot 3, the Client has fetched the details of OEM, this lets the ICA (client) to connect with the OEM-server. The authentication can be seen in Screenshot 4. The Device-id and passwords are given to user by the OEM along with the purchase of the device. After successful-authentication, client will request the IP-address of the device and as a response, remote-host on cloud will share the IP-address of the device. Remember that, the device has a heart-beat, which reports the device IP-address to the remote-host.

**Probe phase:** Once the client has received the IP address of the device, it can directly connect and communicate with device. Now, there is no need to stay connected with the server, so client can close the connection with the OEM-server. After connecting with the device, client will send a PROBE-request packet and as a response, device will send the probe-response (as defined in the "Open OneIoT Protocol"), which will be the manifest-file (JSON-file), as shown in Fig. 12. The Client interprets the JSON-file as per the manifest-grammar and a graphical-user-interface is generated as shown in Screenshot 5 of Fig. 13. By using this GUI, Client can operate the air conditioner.

**Invoke phase:** Once probing is done, the client can invoke operations on the device. Once an operation is invoked, that operation is performed on the device and it will reply with the response of that invoke (as defined in "Open OneIoT Protocol"). For example, in

the AC example, if previously the temperature was 16 °C as in Screenshot 5 of Fig. 13, then the user can now choose to click the increase temperature button (see Screenshot 6) and this will send an invoke-packet to the device raising the temperature to 16.5 °C. This can be seen in Screenshot 8 of Fig. 14. Thus, temperature increase has been effected by 0.5 °C in a single click, because in the main manifest file, the step-size specified is '0.5' (see Fig. 12).

In this section, we took the example of an Air-conditioner, because it has complete characteristics of an IoT device *i.e.*, both sensors and actuators. Thus, the proposed protocol and grammar is completely capable of handling diverse IoT-devices across domains. Hence, this Air-conditioner example demonstrates all interactions between functional entities in the architecture. The entire code-base for this project is made available as an open-source project *via* the repository accessible *via* this hyperlink https://doi.org/10.5281/zenodo.13836351. We intend to benefit from the cybersecurity related advantages associated with open-source project as reported in *Clarke, Dorwin & Nash (2009)* and *Fischer (2005)*.

## SECURITY

In our protocol, the IoT-device, on gaining Internet-access, periodically 'pings' the remote-host (of the OEM). This ensures a constant visibility of the IoT-device from the very instant it comes live online. This is done by our solution's 'tick rate' parameter, which is configurable. The remote-host is hardwired to listen to such 'heartbeats' from Things, because of which the local cache always has the last known address. This address is used as a fallback in the event of an IP-address change. If in case the things IP-address changes, the next heartbeat logs the new address with the host. Sine the client is not yet aware of this address change, it might still reach to the outdated address. In that case, any attempts to invoke a service results in: either (i) a timeout, because no device on that address, or (ii) a protocol failure or an authorisation failure on the event that another IoT-device takes up the old-IP. In the second case, the device fails to understand the protocol or understands the protocol; but cannot authenticate the request due to a key mismatch. These two cases require a call for a cache update on the client side, and the cloud uses the protocol's specific method to update its cache, and retry the request. As we employ salt in the encryption, the key cannot be broken as explained below.

As IoT-devices' microcontroller hardware are very constrained, we need to incorporate security that is affordable by the microcontroller. Since our protocol essentially provide web-based interfaces for physical devices, any exploitation would lead to compromising the associated physical systems. The most affordable security is by encrypting 'raw' data on an end-to-end basis, with accepted encryption standards that provide sufficient encryption entropy. The only fields worth exposing in plain-text would include a header describing the protocol name and version, along with the encryption type used. As mentioned in the "Open OneIoT Architecture", communication would only be possible post successful registration that would distribute an authorization key, post registration for communication. The authorization-key itself could be the device-key, or an OAuth2 like key, which can be revoked as and when needed, to better support multi-client ecosystems

and is recommended. Our protocol can support both these options. In either cases, because of salting, the encryption remains robust against correlation-based attacks. Thus, in designing a secure packet, the presence of a salt (*Morris & Thompson, 1979*) is very important. Consider the scenario where an attacker taps into a private network and sniffs encrypted-packets, which when not salted would lack sufficient entropy allowing the attacker to correlate the encrypted packets (*Chanal & Kakkasageri, 2020*; *Tewari & Gupta, 2020*). For example, salting with a random number generated at the instance of encryption would increase the entropy of the cipher-text by multiple orders of magnitude.

## DISCUSSION AND CONCLUSION

The Open OneIoT platform presented included an application layer IoT protocol that runs on top of an Operating system, which implements the networking stack. The experimentation setup, as illustrated in the previous section, consisted of Raspberry Pi400 board (*McManus & Cook, 2021*) as the device side node, which runs on Raspberry PI OS. The illustrated ICA, called Open OneIoT (O2IoT) app, is implemented for the Android-based smartphones. The manifest is generated by the OEMs of IoT-devices as per the device capabilities adhering to the manifest-grammar as defined in "Manifest Grammar". This grammar makes it possible for automatic generation of the manifest file, which post successful authentication, gets transferred to the O2IoT ICA on the user's mobile phone, as per the defined Open OneIoT protocol phases. The O2IoT app generates the UI on the fly that lets user access the functionalities of the IoT device.

In the experimentation section, the overall manifest file contents and its exchange together with UI generation has been explained. This manifest file is in plain text based JSON-file. If OEMs, due to confidentiality reasons, require that the manifest file to be encrypted, depending on the need, we can go for appropriate encryption supporting hardware, with an appropriate OS, this shall be straight forward. In such a case, the Open OneIoT protocol phases would be pre-fixed with a matching encryption/decryption as necessary. When the internet connectivity fails, then the ICA would act as an external peripheral for the IoT-device. The very generic Open OneIoT reference architecture presented in "Open OneIoT Architecture" resulted in a minimum functionality criteria definition, which makes a device to go online as an IoT-device. This would form as guideline for new vendors to develop IoT devices as per the common minimum criteria.

This article introduced an innovative approach for addressing the challenge of heterogeneity within IoT ecosystems through the implementation of a manifest-file-based device capability-publishing system. This approach enables IoT-devices to publish their capabilities in a standard-format thus leading to a seamless integration with our proposed unified ICA. With the defined manifest-grammar, manifest-files can be validated automatically and an ICA can generate device-specific user interfaces on the fly, providing users with a tailored experience for each connected device.

Moreover, our approach significantly simplifies the user experience by eliminating the need for manual configuration of device interfaces. This is further supported *via* preset-modes saved in sub-manifest-file. Furthermore, the developed ICA as a single app significantly reduces the cognitive load on users.

In conclusion, the integration of manifest file-based device capability publishing *via* our application layer protocol represents a significant step forward in addressing the challenges of heterogeneity within IoT ecosystems. By providing a standardized mechanism for describing device capabilities and dynamically generating device-specific user interfaces, our work would pave the way for a more streamlined, interoperable, and non-overwhelming user-friendly IoT.

### Funding

This work has been supported by a grant awarded by the TiH-IoT, which is established under the aegis of the National Mission on Interdisciplinary Cyber Physical Systems (NMICPS) of the Department of Science and Technology (DST), Government of India. The corresponding document number is TIH-IoT/2023-12/TDP6/Core/SL-005 with Project Code TDP06-A-09. There was no additional external funding received for this study. The funders had no role in study design, data collection and analysis, decision to publish, or preparation of the manuscript.

### Grant Disclosures

The following grant information was disclosed by the authors:
TIH-IoT/2023-12/TDP6/Core/SL-005: TDP06-A-09.

### Competing Interests

The authors declare that they have no competing interests. Krishna Chaitanya Bommakanti is employed by Adonmo. Srinivas K. V. is employed by Motorola Mobility.

### Author Contributions

- Nayancy Gupta conceived and designed the experiments, performed the experiments, analyzed the data, performed the computation work, prepared figures and/or tables, authored or reviewed drafts of the article, and approved the final draft.
- Gourinath Banda conceived and designed the experiments, performed the experiments, analyzed the data, prepared figures and/or tables, authored or reviewed drafts of the article, and approved the final draft.
- Krishna Chaitanya Bommakanti performed the computation work, prepared figures and/or tables, and approved the final draft.
- Venkata Srinivas Kothapalli analyzed the data, authored or reviewed drafts of the article, and approved the final draft.

### Data Availability

The code is available at GitHub: https://github.com/nayancyiiti/IoT.

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
