# Peer review of "A novel IoT-device management platform for on-the-fly generation of user interface via manifest-file addressing heterogeneity"

_PeerJ Computer Science, doi:10.7717/peerj-cs.2480_

## Round 0.1 · original submission · Major Revisions

Two reviewers have provided feedback on this submission. Their views are largely positive, but they have a number of comments, some of which might require a substantial amount of work to address.

Reviewer 1 ·

Basic reporting

The paper is clearly written and well organized. Unambiguous professional English is used, but some typos have to be removed. The introduction states the problem, which is related to diversity of IoT Control Apps (ICAs) and the purpose of the paper, namely to propose a unified way for IoT device management. The significance of this manuscript is explained well. The authors claim the development of a novel manifest-based IoT device platform and an application layer protocol for IoT device management. They provide some technical details, including the manifest grammar and results of an experiment. The security aspects are also discussed. The authors consider the related works in 42 references. The literature is relevant and well referenced. The structure of the manuscript conforms to PeerJ standards. Most of the figures are relevant, labeled and described.

Experimental design

The research is within the scope of the journal. The research question is well defined and is relevant. Methods described with sufficient details. The authors discuss some implementation details of the proposed Open OneIoT protocol with focus on the manifest file concept, the manifest grammar, which is formally described, and the algorithm traversing the trees. An illustration of a main manifest file and the respective JSON description generation is provided. An experiment is setup which simulates the behavior of Clint node, Device node and Cloud node. The different phases of the Open OneIoT protocol are simulated including user login, IoT device registration, probe phase and invoke phase.

Validity of the findings

The reviewer does not detect plagiarism.

Additional comments

General comments
1. The topic is interesting but it is not new one – the considered problem exists since the deployment of IoT solutions and many standardization efforts are done, (Please, refer to ETSI M2M specifications and OMA specifications, i.e. ETSI TS 102 690). In this context, the most important issue is why the authors do not consider the works of Open Mobile Alliance (OMA) Device Management which provides a standardized framework for management of devices including IoT devices. The standard provides a detailed device description from the management point of view including JSON descriptions. The authors must consider the OMA and ETSI works on M2M communications and to justify how their method and protocol is better than these works.
2. The next important issue is regarded to the manifest file. The proposed IoT description is too general and does not cover many of the specific aspects of the IoT devices. For example, the missing device information is the SW version and HW version. The manifest grammar does not include the IoT device actuator capability. Furthermore, in addition to sensors and/or actuators, the device may contain also other capabilities such as an external storage, communication modules, display, etc., which are not considered. Further, the capability can be also manipulated (on or off) and the required actions have to be defied. The devices description needs to provide information about the batteries of a device, device memory, trap events and trap event actions, performance log and the respective action on it, etc.
3. As a part of the IoT device description, each part of the device needs to have subscriptions information, which enables notify the subscriber about any changes. Such information in the manifest file is missing.
4. The authors claim to define Open OneIoT architecture illustrated in Figure 1. Usually, when an architecture is defined, in addition to description of functional entities, it is also provided a description of interface functionality which is missing in the manuscript. Figure 1 is not correct one. Where is the remote host which runs on the cloud? The network connectivity must mot be illustrated like a entity as it exist between user agents and the cloud and between IoT nodes and the user agents.
5. When a protocol is described the protocol operations are illustrated by sequence diagrams which are missing in the manuscript. Each of the protocol phases must be illustrated. In addition, sequence diagrams must be provided for information retrieval, subscriptions and notifications.
6. Figure 2 and Figure 3 are low informative. Furthermore, there must be included additional information in the protocol message format as the addresses of the source and destination, timestamp, etc.
7. The Open OneIoT protocol is pretended to be an application protocol and the authors should provide the protocol stack used.
8. The manifest grammar, provided on two pages, even incomplete, is better to be illustrated by figures describing the parts of the device descriptions (please refer to ETSI TS 102 690).
9. In the related works, the authors need to discuss different methods for IoT control and also other navigating IoT communication protocols except the considered ones (OMA device management protocol, OMA data distribution service, etc.).
10. The figure quality could be improved e.g. figure 7 and figure 8, showing grammar trees are spread over a full page and could be resized to take up less space without reducing the size of the text in them.
11. There are some typos that must be removed (e.g. line 137,213, 226 etc.)

Comments on references (The reviewer declares that he/she is not an author or co-author of any of the works recommended below!)
1. References with regard to IoT architecture: the cited references even adequate are out of date, it is better to refer to more up-to-date surveys such as https://doi.org/10.1016/j.iot.2022.100626, https://doi.org/10.1007/978-3-030-70713-2_58
2. References with regard to IoT device management: The cited references are out of date, it is better to refer to more up-to-date ones such as: doi: 10.1109/WPMC48795.2019.9096203, doi: 10.1109/TrustCom60117.2023.00256, doi: 10.1109/ICDT61202.2024.10488939
3. Suggested references with regard to IoT device heterogeneity: doi: 10.1109/ACCESS.2020.3039368, doi: 10.1109/JIOT.2022.3221967
4. Suggested references with regard to Cloud based IoT device control: doi: 10.1109/IoT60973.2023.10365372, doi: 10.1109/ACCESS.2022.3141977, doi: 10.1109/TDSC.2022.3204720

Annotated reviews are not available for download in order to protect the identity of reviewers who chose to remain anonymous.

Reviewer 2 ·

Basic reporting

The manuscript is written in clear and professional English. However, certain sections could benefit from further editing to improve readability and clarity. For instance, the introductory sentences are lengthy and could be broken down into shorter, more digestible parts.
The introduction provides a solid context, outlining the challenges in the IoT domain related to device heterogeneity and the proliferation of multiple control apps. The problem statement is clear, and the significance of the research is well-justified. However, the background could be enhanced by discussing more recent advancements in IoT device management and their limitations.
The literature is well-referenced and relevant. The paper discusses various related works and protocols, including CoAP, MQTT, and the Matter protocol, providing a comprehensive background. However, the authors could improve the depth of the comparison with existing solutions to highlight the novelty of their approach more effectively.
The structure of the manuscript generally conforms to academic standards, with a logical flow from the introduction to the conclusion. However, the text contains occasional formatting issues, such as inconsistent font sizes and spacing, which should be addressed for a more polished presentation.
The figures included are relevant and help in understanding the proposed platform.
The manuscript does not mention raw data. The authors should ensure that all relevant data and supplementary materials are accessible and properly referenced according to PeerJ's guidelines.

Experimental design

The research question is well-defined, focusing on addressing the heterogeneity in IoT devices through a manifest-file-based platform. The significance of the research is well-articulated, and the study aims to fill a clear gap in the current literature. The methods are described in detail, allowing for reproducibility. The development of the manifest grammar and the OneIoT protocol is well-documented, and the inclusion of an application-layer protocol to handle device communication is innovative. However, the paper could benefit from a more explicit discussion on the selection of test cases and scenarios used in the experimentation section.

Validity of the findings

The results provided in the paper are robust and the analysis has been carried out using appropriate methods. The authors have performed a thorough evaluation of their platform, demonstrating its effectiveness in managing diverse IoT devices. However, more extensive testing under different conditions, such as varying network environments or device types, would strengthen the validity of the findings.
The conclusions are well-supported by the data presented. The authors effectively link their results back to the original research question, demonstrating that the OneIoT platform can indeed manage multiple IoT devices through a single application interface. However, the paper would benefit from a more detailed discussion of potential limitations and areas for future research.

Additional comments

The main strengths of the manuscript lie in its innovative approach to solving the problem of IoT device heterogeneity through a manifest-based platform. The solution is practical and has the potential for significant impact in the IoT field, particularly in reducing the complexity of device management for end-users.
One of the main weaknesses is the lack of extensive real-world testing. While the initial results are promising, the platform's performance in a broader range of scenarios needs to be validated.

---

## Round 0.2 · accepted · Accept

The reviewers believe that all their comments have been addressed satisfactorily.

Reviewer 2 ·

Basic reporting

The manuscript is written in clear and professional English. However, certain sections could benefit from further editing to improve readability and clarity. For instance, the introductory sentences are lengthy and could be broken down into shorter, more digestible parts.
The introduction provides a solid context, outlining the challenges in the IoT domain related to device heterogeneity and the proliferation of multiple control apps. The problem statement is clear, and the significance of the research is well-justified. However, the background could be enhanced by discussing more recent advancements in IoT device management and their limitations.
The literature is well-referenced and relevant. The paper discusses various related works and protocols, including CoAP, MQTT, and the Matter protocol, providing a comprehensive background. However, the authors could improve the depth of the comparison with existing solutions to highlight the novelty of their approach more effectively.
The structure of the manuscript generally conforms to academic standards, with a logical flow from the introduction to the conclusion.
The figures included are relevant and help in understanding the proposed platform.

Experimental design

The research question is well-defined, focusing on addressing the heterogeneity in IoT devices through a manifest-file-based platform. The significance of the research is well-articulated, and the study aims to fill a clear gap in the current literature. The methods are described in detail, allowing for reproducibility. The development of the manifest grammar and the OneIoT protocol is well-documented, and the inclusion of an application-layer protocol to handle device communication is innovative.

Validity of the findings

The results provided in the paper are robust and the analysis has been carried out using appropriate methods. The authors have performed a thorough evaluation of their platform, demonstrating its effectiveness in managing diverse IoT devices. However, more extensive testing under different conditions, such as varying network environments or device types, would strengthen the validity of the findings.
The conclusions are well-supported by the data presented. The authors effectively link their results back to the original research question, demonstrating that the OneIoT platform can indeed manage multiple IoT devices through a single application interface.

Additional comments

The main strengths of the manuscript lie in its innovative approach to solving the problem of IoT device heterogeneity through a manifest-based platform. The solution is practical and has the potential for significant impact in the IoT field, particularly in reducing the complexity of device management for end-users.
The authors have addressed the issues arisen during the previous round of review.